# Evaluation of the Burden of HPV-Related Hospitalizations as a Useful Tool to Increase Awareness: 2007–2017 Data from the Sicilian Hospital Discharge Records

**DOI:** 10.3390/vaccines8010047

**Published:** 2020-01-26

**Authors:** Vincenzo Restivo, Claudio Costantino, Livia Amato, Stefania Candiloro, Alessandra Casuccio, Marianna Maranto, Alessandro Marrella, Sara Palmeri, Stefano Pizzo, Francesco Vitale, Emanuele Amodio

**Affiliations:** Department of Health Promotion, Maternal and Infant Care, Internal Medicine and Medical Specialties, “G. D’Alessandro”, University of Palermo, Via del Vespro 133, 90127 Palermo, Italy; claudio.costantino01@unipa.it (C.C.); livia.amato@unipa.it (L.A.); alessandra.casuccio@unipa.it (A.C.); mariannamaranto@libero.it (M.M.); alessandro.marrella@unipa.it (A.M.); sarapalmeri@libero.it (S.P.); stefano.pizzo@unipa.it (S.P.); francesco.vitale@unipa.it (F.V.); emanuele.amodio@unipa.it (E.A.)

**Keywords:** HPV-related disease, cervical cancer, burden, hospitalization rate, HPV vaccine, cervical cancer screening, impact evaluation, Italy, vaccine effectiveness, screening effectiveness

## Abstract

In light of the implementation of human papillomavirus (HPV) prevention strategies, epidemiological studies in different geographical areas are required in order to assess the impact of HPV-related diseases. The purpose of the present study was to describe the burden of HPV-related hospitalizations in Sicily. A retrospective observational study estimated 43,531 hospitalizations attributable to HPV from 2007 to 2017. During the observed period, there was a decrease for all HPV-related conditions with a higher reduction, among neoplasms, for cervical cancer (annual percent change (APC) = −9.9%, *p* < 0.001). The median age for cervical cancer was 45 years old, with an increasing value from 43 to 47 years (*p* < 0.001). The age classes with greater decreases in hospital admissions for invasive cancers were women aged 35 years or more (APC range from −5.5 to −9.86) and 25–34 years old (APC = −11.87, *p* < 0.001) for women with cervical carcinoma in situ. After ten years for vaccine introduction and sixteen years for cervical cancer screening availability, a relatively large decrease in hospital admissions for cervical cancer and other HPV-related diseases in Sicily was observed. Some clinical characteristics of hospitalization, such as increasing age, are suggestive clues for the impact of preventive strategies, but further research is needed to confirm this relationship.

## 1. Introduction

Human papillomavirus (HPV) is considered the most common causative agent responsible for sexually transmitted infections in the general population [1]. The prevalence of HPV infection varies greatly around the world, being dependent on the target population and the severity of the disease [2]. HPV serotypes can be classified into high- and low-risk types according to their capacity to induce cancer [3]. The low-risk types HPV 6 and HPV 11 cause 90% of anogenital warts and benign/low-grade abnormalities in the genital areas [4], and they are also responsible for causing recurrent respiratory papillomatosis, a proliferative disease of the upper aero-digestive tract [5]. The carcinogenic types are responsible for up to 4.5% (630,000 cases) of all new cancer cases (8.6% in females; 0.9% in males) and cause 29.5% of all HPV-related cancers [6]. In detail, HPV 16 and 18 cause 70% of cervical cancers, a relevant proportion of vaginal (78%), anal (88%), as well as some vulvar (25%) or penile (50%) cancers and, together with HPV types 31/33/45/52/58, are responsible for about 90% of HPV-related cancers worldwide [7].

Cervical cancer remains the fourth most common female cancer globally, with an estimated 570,000 cases and 311,000 deaths yearly [8]. According to the most recent data, every year in Italy there are 6,500 cancer cases of HPV-related new diagnoses with a higher burden of cervical cancer (45%) [9]. In light of the recent development of HPV-type-specific prophylactic vaccines and implementation of cervical cancer screening [10], epidemiological studies in different geographical areas are required in order to assess the impact of HPV infections. Australia was the first country to roll out an extensive vaccination program for all women aged 12–26 years from 2007, reaching a high vaccination coverage in a short time. After the introduction of the vaccination program, cytology and histopathology data showed a 42% decrease in the incidence of high-grade cervical lesions in girls younger than 18 years [11]. A more rapid instrument to estimate the burden of HPV-related diseases could be the hospitalization records related to genital warts, cervical cancer and anal and penile neoplasms [12]. In Sicily, anti-HPV vaccination has been offered since 2008 to all girls in their 12th year of life (birth cohort 1996), and since 2015 it has also been offered to males of the same age (birth cohort 2003) [13]. Furthermore, it has been offered in co-payment to women up to 45 years old and to men up to 26 years old [12]. In 2017, the nine-valent vaccine replaced the quadrivalent formulation for girls and boys aged 12 years [14]. 

The other preventive practice useful to reduce the risk of mortality for cervical cancer is cervical screening. A meta-analysis of 12 case-control studies demonstrated that it is strongly associated with a decreased risk of invasive cervical cancer (OR 0.35, 95% CI 0.30–0.41) [15]. Large-scale clinical trials suggest that primary HPV DNA screening is more effective at preventing cervical cancer than screening with cytology at shorter intervals [16]. In Italy, the implementation of organized cervical screening programs has been recommended since 1996 [17]. Moreover, it has been included as an essential level of care since 2001, which implies free screening availability for all Sicilian women aged 25–65 years [18].

The purpose of the present study was to describe the burden of HPV-related hospitalizations in Sicily during 2007–2017, after ten years of vaccine implementation and sixteen years of screening availability.

## 2. Materials and Methods 

A retrospective observational study was conducted using Sicilian hospital discharge records (HDRs) collected from 2007 to 2017. Sicily is the fourth most populous Italian administrative region, with 5,026,989 inhabitants (2018 Census). The Sicilian Hospital Admission Database is the comprehensive database of all hospital discharges observed in public and private regional health care facilities. It provides information on hospitalized patients, notably demographic (e.g., sex, age, gender and city of residence) and medical data (start and end of stay, length of stay, reasons for hospital admission and discharge diagnosis, medical unit of stay, medical procedures performed during the stay, and, where applicable, hospital inpatient date of death) [19]. It also contains economic information for each hospital stay. All information concerning the same patient is linked together using a unique identifier. In all Italian administrative regions, such as Sicily, HDRs are codified accordingly to the ICD-9-CM coding system. In order to analyze HPV-related diseases, the following diagnostic codes in one of the six diagnostic positions were used: 078.11 (condyloma acuminatum), 140.0–149.9, 195.0, 230.0, 235.1 (head and neck cancers), 154.2–154.8, 230.5–230.6 (anal cancers), 180.0–180.9, 233.1, 622.1, 654.6, 795.0–795.1 (cervical cancers), 184.0–184.8 (genitourinary tract cancers: vagina, labia, clitoris) and 187.1–187.9, 233.5 (penile cancers). Furthermore, in order to improve the sensitivity of HPV hospitalization burden estimation, the following ICD-9-CM intervention codes were evaluated: 67.2 (conization of the cervix), 67.32 (demolition of the cervical lesion by cauterization), 67.33 (demolition of the cervical lesion by cryosurgery). Each hospitalization was potentially attributed to HPV if it had at least one ICD-9-CM diagnostic code or one ICD-9-CM intervention code in any diagnostic or procedural position.

According to several published studies, the following attributable fractions (AFs) were used to quantify the proportion of hospitalizations attributable to HPV: 88% of anal cancers, 50% of penile cancers, 26% of head and neck cancers, 77% of female genitourinary tract cancers, 100% for cervix cancer and warts [20]. In depth, for each evaluated pathology, HPV-attributable hospitalizations were calculated by multiplying the specific AF by the total number of hospitalizations for that condition according to the following formula:

Attributable hospitalizations for each HPV-related disease = Specific attributable fraction * Total hospitalizations for each HPV-related disease.

Quality control for duplication of hospital admissions and hospitalizations with missing data was performed. The following data were extracted, as de-identified data, for each selected hospital admission: sex, age, days of hospital stay and country of origin. 

Data are described using absolute and relative frequencies for categorical variables, and mean ± standard deviation (SD) or median with interquartile ranges (IQR) for quantitative variables, depending on normal or not normal distribution. The hospitalization rate was calculated by dividing the number of hospital admissions by the total number of at-risk Sicilian population in each observed year [21]. The hospitalization rate was also calculated per 100,000 persons based on the Sicilian population size for each 10-year age group. For quantitative variables, the Mann–Whitney test was used to evaluate significant differences. Statistical analyses were performed using STATA v14.2 software (StataCorp LLC, College Station, TX, USA). The Joinpoint model (Joinpoint version 4.6.0.0, 2018) was used to evaluate the time trends of standardized rates, the direction and the intensity of the (linear) trend, and the average annual percent change (APC). APC has to be considered a summary measure of the trend over a given fixed time interval that is computed as a weighted average of the annual percent change emerging from the joinpoint model, using weights equating to the length of the period interval. The final model is based on linear segments connected at joinpoints that represent the best fit of observed data. For all analyses, a *p*-value ≤ 0.05 was assumed to indicate statistical significance (two-tailed).

## 3. Results

Overall, 73,284 hospitalizations for potential HPV-related diseases were detected among Sicilian people during the entire period. According to the reported attributable fractions, it was estimated that 43,531 hospitalizations could be considered related to HPV in Sicily. As reported in Table 1, the hospitalization rate for cancer of the cervix was 106.3 per 100,000 person-years, followed by warts (14.7 per 100,000 person-years) and oropharynx cancer (7.4 per 100,000 person-years). Analyzing gender distribution, a higher prevalence of cervical cancer was observed among women, while in men, head and neck cancer (43%) had higher prevalence (data not shown in table). For all analyzed HPV-related diseases, decreasing hospitalization rates were observed from 2007 to 2017. Only penis cancer peaked in 2009 and subsequently decreased over the observation period. 

Figure 1 shows the hospitalization rate trends among HPV-related diseases in Sicily. Overall, there was a decrease for all HPV-related conditions. Genital warts was the disease with the highest decrease in hospital admissions from 27.3 to 6.4 per 100,000 per year (APC = −12.5%, *p* < 0.001). Among neoplastic diseases, the greatest reduction was detected for cervical cancer, which went down from 166.6 to 63.3 per 100,000 per year (APC = −9.9%, *p* < 0.001), followed by oropharynx cancer from 10.0 to 4.7 per 100,000 per year (APC = −8.7%, *p* < 0.001), and penis cancer with a hospitalization rate decrease from 3.2 to 2.2 per 100,000 per year (APC = −5.2%, *p* = 0.005).

Among the cancers, the most common was cervical cancer with 30,430 hospitalizations occurring in 23,049 women. The demographic and clinical characteristics of hospitalizations with cervical cancer are reported in Table 2. Women with a hospital admission for cervical cancer were most frequently Italian (95.5%), followed by Romanian (2.6%) and Polish (0.3%). Furthermore, 91 women (0.3%) hospitalized for cervical cancer died during their hospital stay. The median hospital stay for cervical cancer was 4 days, with an increasing value with time from 4 to 5 days of hospital stay (*p* < 0.001). The median age was 45 years old, with an increasing value from 43 to 47 years (*p* < 0.001). 

The number of patients with a hospital admission for invasive cervical cancer was 6,166 (20.3%). Among these, as shown in Figure 2, the most prevalent age class was 45–54-year-old women (26.7%) followed by 55–64 (22.7%) and 35–44 (19.3%). Hospitalization rates showed decreasing trends from 2007 to 2017 for age groups aged over 34 years old (APC range from −5.5 to −9.86).

On the other hand, people with cervical carcinoma in situ (CIS) had the greatest cervical cancer burden with 24,264 (79.7%) hospital admissions. As reported in Figure 3, the age class most represented for CIS was 35–44-year-olds (29.8%) followed by 45–54 (26.1%) and 25–34 (20.8%). From 2007 to 2017, greater decreases were reported among 25–34-year-olds (APC = −11.87, *p* < 0.001) followed by 35–44 (APC = −10.88, *p* < 0.001).

## 4. Discussion

This observational study showed the burden of HPV-related disease in Sicily from 2007 to 2017. After ten years for vaccine introduction and sixteen years for cervical cancer screening availability, a relatively large decrease in hospital admissions for cervical cancer and other HPV-related diseases was observed. 

The reduction in hospitalization rates could be attributable to several factors, including HPV vaccination, which reduces HPV infection and carcinogenesis, cervical screening programs, which are well known to reduce the incidence and mortality of cervical cancer, and level of adherence to preventive practices (such as women’s compliance and attendance over time, as recommended, for their next screening test or for their next vaccine dose).

Data reported in Sicily can be compared with another Italian study conducted from 2001 to 2012 using a similar methodology [20]. According to this latter study, cervical intraepithelial neoplasia (CIN) III and cervical cancer were the HPV-related pathologies with the highest impact on Italian populations. The decrease in cervical cancer burden (64%) observed in Sicily was higher than in Italy as a whole (24%). Furthermore, the Sicilian burden decrease was higher than that reported in Spain, where there was a decrease of 23% [12]. The lower decrease in hospital admissions observed in the other cited studies can be explained at least by considering the different years in which preventive strategies were implemented. In detail, the Spanish study analyzed only one year after vaccine implementation and the Italian study 5 years after introduction, while the Sicilian study considered a ten-year period after the beginning of the immunization campaign.

At the Italian administrative region level, there are two different studies that described the burden of HPV-related hospitalization in Tuscany and Veneto in relation with the different adopted vaccination strategies [22,23]. These studies both showed a significant reduction over time in hospitalizations for vulvar/vaginal warts, especially for patients from 12 to 20 years old, the age group for which the overall HPV vaccination coverage had reached about 75%. These data suggest an early decline in the rate of HPV-related hospital admissions in a population with a higher HPV vaccination coverage [24].

Among Sicilian women, there was an increase in age at hospitalization that could be associated with the implementation of HPV vaccination and screening strategies. In this sense, after the implementation of preventive strategies in a target age class (12 years old for vaccination and 25 years old for HPV screening), it was demonstrated that the mean age of the cohort of susceptible people increases year by year [25]. Furthermore, an increase in length of hospital stay was observed in HPV-related Sicilian hospital admissions. A potential explanation for this finding could be that hospitalization is mostly needed for invasive cervical cancer, as women suffering from this disease usually undergo a radical hysterectomy and/or lymphadenectomy. Preventive strategies are tailored for adolescents (HPV vaccine) or young adult women (cervical cancer screening). Therefore, a higher likelihood of hospital admission for severe staging of cervical cancer due to an intensive surgical treatment with a longer hospital stay is expected for older adult women without vaccination or screening. Moreover, the increased age of hospitalized Sicilian women may increase the length of hospital stay due to the presence of comorbidities that are more prevalent among older women [26]. 

Sicily was the principal entrance for migrant populations from Africa and the Middle East to Europe [27,28]. Several analyses found that immigration might delay the timeline to elimination of cervical cancer, assuming that migrants do not benefit from HPV vaccination [29]. Notwithstanding, the HPV burden among the migrant population was low in Sicily. Consequently, many immigrants could have benefited from HPV vaccination programs in their countries of birth, or they arrive at an age that is young enough for them to be eligible to receive free HPV vaccination in Sicily. Some immigrants might arrive with an infection that subsequently causes cancer, since these are typically acquired at younger ages, which reinforces the importance of screening [30].

This study highlights a greater hospitalization decrease for CIS among 15–24-year-old women who had access to anti-HPV vaccination. This reduction could be a clue for HPV vaccine effectiveness, as reported by several studies. The most updated data were reported in the meta-analysis of Drolet et al. including 65 studies from 14 high-income countries, pooling data from over 60 million individuals for up to eight years of post-vaccination follow-up [31]. The report shows overall significant reductions of 80% in the prevalence of types HPV 16 and 18, a reduction of 70% in anogenital wart diagnoses, and a significant reduction in CIN2+. However, only a few studies reported the impact of HPV vaccines on invasive cervical cancer due to high latency periods, and probably more years could be needed for observing significant differences. Preliminary data are available from the follow-up of cancer registries of Finnish women vaccinated in HPV-vaccine clinical trials, which linked data from vaccination trial cohorts and compared them to non-vaccinated cohorts [32]. No cancer case was diagnosed in the vaccinated cohort, versus eight cases of cervical cancer, one vulvar cancer, and one oropharyngeal cancer in the unvaccinated cohort. That was equivalent to 6.4 cases per 100,000 person-years among unvaccinated women in comparison to a reduction of 100% among the vaccinated ones. 

In the Sicilian study, a greater decrease in invasive cervical cancer disease was recorded among women older than 35 years who had access to cervical cancer screening. This result was in accordance with what was reported in Scotland, where vaccination and screening records of nine birth cohorts of 20-year-old women were extracted to evaluate the effectiveness of the HPV vaccine in preventing cervical precancerous lesions [33]. The analysis identified 89% reduction of prevalent CIN3+; 88% reduction of CIN2+; and 79% reduction of CIN1 in vaccinated women born in 1995 and 1996 compared with unvaccinated women born in 1988.

Despite various evidence for anti-HPV vaccine effectiveness, Sicilian vaccination coverage still remains low, with a mean level of 51%, and far from the minimum value required by the Italian Ministry of Health [13]. Furthermore, the screening practice is still far from optimal levels, reaching an adherence of 66% among women from 25 to 65 years old in Sicily [34]. As reported in Australia, higher vaccination coverage and screening adherence for a longer time are needed to eliminate HPV circulation [35]. The Australian estimates were based on a 5-year period screening coverage of 83% from 18 to 69 years and a vaccination coverage of 82% in girls and 76% in boys aged 12 years. The Australian Health Authorities predicted reaching the rare cancer threshold of six new cases per 100,000 women each year in 2020 and cervical cancer elimination (defined as four new cases per 100,000 women each year) from 2028.

Consequently, the Sicilian data could be useful to build up evidence for healthcare authorities to know trends of HPV-related disease [36]. Another critical topic is to build up an adequate formative strategy for all healthcare authorities involved in HPV vaccination counseling (gynecologist, general practitioner, pediatrician and public health physician), in order to standardize the communication process for the general population and to reach high-risk groups [37]. Several studies conducted in Sicily detected factors associated with lower adherence to HPV vaccination and HPV screening, including: higher education level, lower participation at school seminars on HPV, lower perception of benefits for HPV vaccination [38], general practitioners’ advice, and perceived susceptibility to cervical disease for Pap test [34]. Only by implementing tailored preventive campaigns for the hard to reach people alongside an increase in preventive service performance can it be plausible to eliminate HPV-related disease in the near future [39].

This study, however, has to be interpreted in light of certain limitations due to the nature of the source of HDRs. Indeed, errors in diagnosis coding would influence the epidemiological data; however, the errors are expected to be very rare for these well-known diseases. As a second limitation, considering the absence of biological results in the HDR, we could not determine the effective part of HPV-related cancers among identified cases. Another limitation could be related to the decrease in HDRs for some procedures that Italian legislation has shifted from hospitalization to ambulatory care [40]. Finally, in our statistical analysis we reported only APC, omitting to describe changes in trends during the pre-specified fixed time interval. The reason for such a choice was due to the fact that, when different intervals’ trends were identified over the period, the slope coefficients for segments in the range of years were relatively similar. Moreover, we were interested in identifying a single summary measure over the entire interval time period for each disease.

However, the main objective of the study was to assess the epidemiology and burden of potentially HPV-related diseases during a long-lasting period, where all these limitations can be weakened. 

## 5. Conclusions

HPV-related burden had a constant decrease in Sicily and this decline could be, at least in part, attributed to ten years of HPV vaccine implementation and sixteen years of cervical screening availability. Some clinical characteristics of hospitalizations, such as increasing age, are suggestive clues for the impact of preventive strategies, but further research is needed to confirm the relationship. These data could be useful to local health authorities for improving vaccination coverage and screening adherence among Sicilian populations using an evidence-based approach in order to evaluate their effectiveness.

## Figures and Tables

**Figure 1 vaccines-08-00047-f001:**
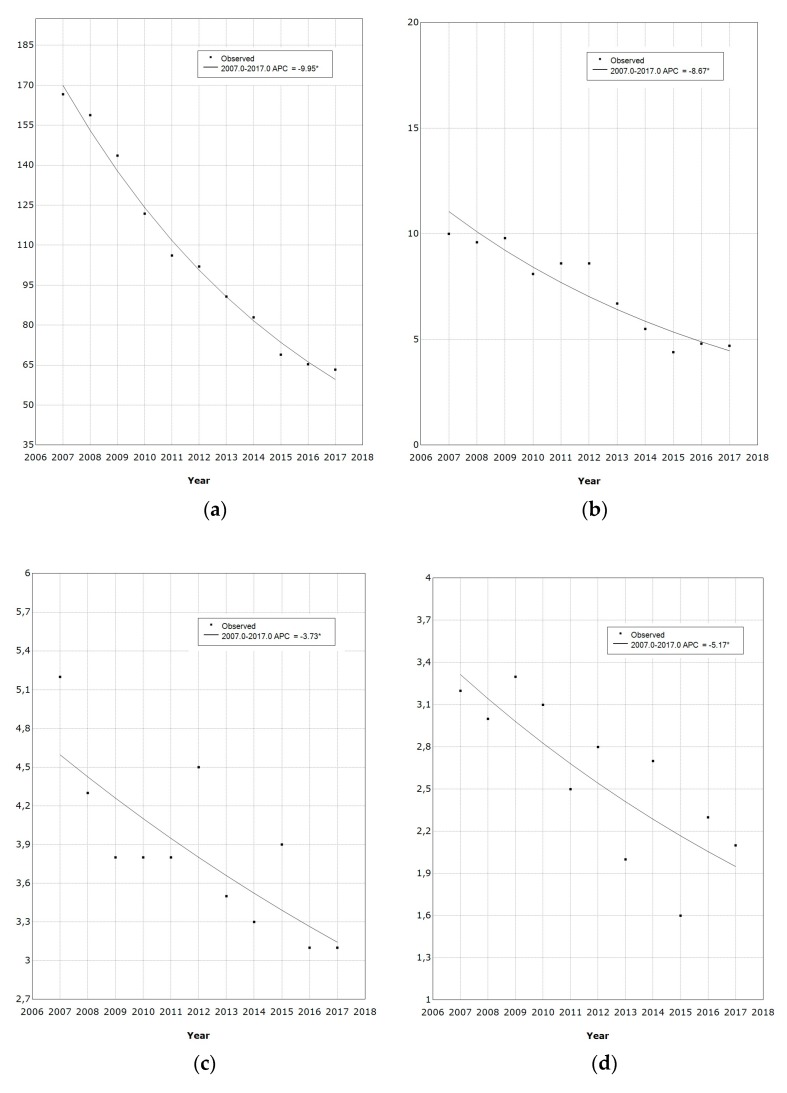
Trends and average annual percentage change (APC) observed for HPV-related hospitalization rates from 2007 to 2017. (**a**) Cervical cancer; (**b**) Oropharynx cancer; (**c**) Anus cancer; (**d**) Penis cancer; (**e**) Vulva and vagina cancer; (**f**) Genital warts. * Indicates that the APC is significantly different from zero at the alpha = 0.05 level.

**Figure 2 vaccines-08-00047-f002:**
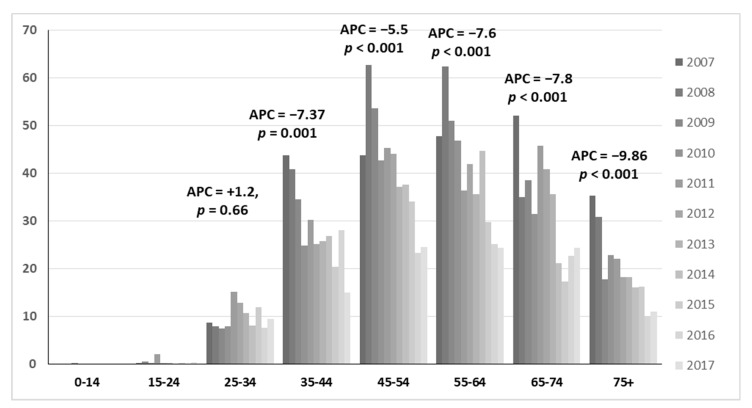
Age distribution of hospital admission rates for invasive cervical cancer from 2007 to 2017.

**Figure 3 vaccines-08-00047-f003:**
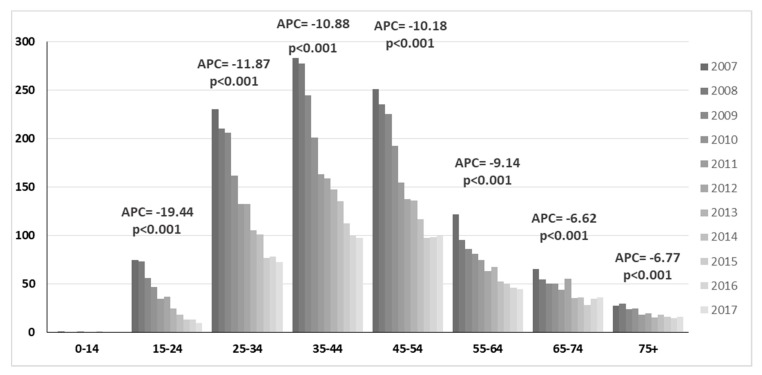
Age distribution of hospital admission rates for cervical carcinoma in situ from 2007 to 2017.

**Table 1 vaccines-08-00047-t001:** Hospitalization rates (per 100,000 person-years) of human papillomavirus (HPV)-related diseases in Sicily.

Year	Cervical Cancer Rate (SD)	Oropharynx Rate (SD)	Anus Rate (SD)	Penis Rate (SD)	Vulva and Vagina Rate (SD)	Genital Warts Rate (SD)
2007	166.6 (2.5)	10.0 (0.4)	5.2 (0.3)	3.2 (0.4)	9.1 (0.6)	27.3 (1.0)
2008	158.8 (2.5)	9.6 (0.4)	4.3 (0.3)	3.0 (0.3)	7.7 (0.5)	26.2 (1.0)
2009	143.6 (2.3)	9.8 (0.4)	3.8 (0.3)	3.3 (0.4)	8.0 (0.5)	19.2 (0.9)
2010	121.8 (2.2)	8.1 (0.4)	3.8 (0.3)	3.1 (0.3)	7.1 (0.5)	14.5 (0.7)
2011	106.1 (2.0)	8.6 (0.4)	3.9 (0.3)	2.5 (0.3)	7.1 (0.5)	12.8 (0.7)
2012	102.0 (2.0)	8.6 (0.4)	4.5 (0.3)	2.9 (0.3)	7.2 (0.5)	13.9 (0.7)
2013	90.7 (1.9)	6.7 (0.4)	3.6 (0.3)	2.0 (0.3)	6.8 (0.5)	13.9 (0.7)
2014	82.9 (1.8)	5.6 (0.3)	3.3 (0.2)	2.7 (0.3)	5.8 (0.5)	9.6 (0.6)
2015	68.9 (1.6)	4.4 (0.3)	3.9 (0.3)	1.7 (0.3)	6.1 (0.5)	8.3 (0.6)
2016	65.3 (1.6)	4.9 (0.3)	3.1 (0.2)	2.3 (0.3)	6.2 (0.5)	9.2 (0.6)
2017	63.3 (1.6)	4.7 (0.3)	3.1 (0.2)	2.2 (0.3)	5.0 (0.4)	6.4 (0.6)
Total	106.3 (0.6)	7.4 (0.1)	3.8 (0.1)	2.6 (0.1)	6.9 (0.1)	14.7 (0.2)

**Table 2 vaccines-08-00047-t002:** Age and length of hospital stay observed for hospitalizations that occurred in patients with cervical cancers.

Hospital Admissions Year	Age, Median (IQR)	*p*	Hospital Stay, Median (IQR)	*p*
2007	43 (34–52)	< 0.001	4 (2–6)	< 0.001
2008	43 (35–52)		3 (2–7)	
2009	44 (35–52)		3 (2–7)	
2010	44 (36–54)		4 (2–8)	
2011	45 (35–54)		4 (2–7)	
2012	45 (36–55)		4 (2–)	
2013	46 (37–55)		4 (2–8)	
2014	46 (37–55)		4 (2–9)	
2015	46 (38–55)		5 (3–8)	
2016	46 (37–56)		5 (3–9)	
2017	47 (38–57)		5 (3–9)	

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
