# Peer review of "Evaluation of the Burden of HPV-Related Hospitalizations as a Useful Tool to Increase Awareness: 2007–2017 Data from the Sicilian Hospital Discharge Records"

_vaccines, 2020, doi:10.3390/vaccines8010047_

Round 1

Reviewer 1 Report

This is an observational study based on medical records data and described the rate change in hospital admission due to HPV-related conditions in Sicily area. Please see my comments as followed:

In the material and methods page 3 line 121, it is not clear how the attributable fractions were used to calculate the fractions attributable to HPV, please clarify the specific statistical methods that were used in this calculation because this calculation produced the core results in this study. Also, the methods part should be better developed. In results page 4 line 147, it may not be appropriate to describe this decline as constant, because rate in this decline is varying in different years. The results should be better described. E.g., on page 6 line 174, it’s not clear to describe it as “distribution”, this in fact is a proportion.

Author Response

Reviewer #1:

In the material and methods page 3 line 121, it is not clear how the attributable fractions were used to calculate the fractions attributable to HPV, please clarify the specific statistical methods that were used in this calculation because this calculation produced the core results in this study. Also, the methods part should be better developed. In results page 4 line 147, it may not be appropriate to describe this decline as constant, because rate in this decline is varying in different years. The results should be better described. E.g., on page 6 line 174, it’s not clear to describe it as “distribution”, this in fact is a proportion.

Answer:

Dear Reviewer,

Thank you for revising our manuscript and for your comments and suggestions. Below you will find a point by point answer to your questions.

Question: It is not clear how the attributable fractions were used to calculate the fractions attributable to HPV, please clarify the specific statistical methods that were used in this calculation because this calculation produced the core results in this study.

Answer: According to the methodology proposed by other authors (see de Martel et al Int J Cancer. 2017 Aug 15; 141(4): 664–670; Baldo et al. BMC Infect Dis. 2013; 13: 462) we have used HPV attributable fractions obtained from the international literature. Using these attributable fractions, for each evaluated pathology, HPV attributable hospitalizations were calculated by multiplying the specific attributable fraction (AF) by the total number of hospitalizations for that condition according to the following formula:

Attributable hospitalizations for each HPV related disease= Specific attributable fraction * Total hospitalizations for each HPV related disease

We have added these last considerations to the material and methods section of the manuscript.

Q: The methods part should be better developed.

A: We have revised the method section and several changes have been made according to suggestions from you and the other Reviewers.

Q: In results page 4 line 147, it may not be appropriate to describe this decline as constant, because rate in this decline is varying in different years. The results should be better described. E.g., on page 6 line 174, it’s not clear to describe it as “distribution”, this in fact is a proportion.

A: We apologize for our inaccuracy about the trend. The sentence has been rewritten accordingly and the term “constant” has been erased. The sentence at page 6 line 174 has been changed. Results have been revised by considering your suggestions and those made by the other Reviewers.  

Reviewer 2 Report

I was invited to review the paper entitled "Evaluation of the burden of HPV-related hospitalizations as useful tool to increase awareness: 2007-2017 data from the Sicilian hospital discharge records". I want to congratulate with Authors for the excellent work. The introduction explain deeply the background of the study. Methods are adequate and results are well presented. In my opinion, this work is very interesting for the field and it improves knowledge about the burden of HPV related hospitalization in a large study period. 

I have only one question: why did you calculated hospitalization rates dividing by the total number of Sicilian population in each observed year? Probably an indirect standardization methods, dividing by the number of population in the first year of observation can provide comparability of each years rate.

Author Response

Reviewer #2:

I was invited to review the paper entitled "Evaluation of the burden of HPV-related hospitalizations as useful tool to increase awareness: 2007-2017 data from the Sicilian hospital discharge records". I want to congratulate with Authors for the excellent work. The introduction explain deeply the background of the study. Methods are adequate and results are well presented. In my opinion, this work is very interesting for the field and it improves knowledge about the burden of HPV related hospitalization in a large study period.

I have only one question: why did you calculated hospitalization rates dividing by the total number of Sicilian population in each observed year? Probably an indirect standardization methods, dividing by the number of population in the first year of observation can provide comparability of each years rate.

Answer:

Dear Reviewer,

Thank you for revising our manuscript and for your appreciation that highlights the importance of population studies evaluating the burden of vaccine preventable diseases. About your question, as you have stated, we have calculated hospitalization rates by dividing hospitalized cases by the total number of at risk Sicilian population. We have preferred this approach also in consideration that during the observation period the demographic structure of the Sicilian population has not changed and thus the results would be not be different than those obtained dividing by the number of population in the first year of observation through an indirect standardization approach. However, if required we will reconsider this approach and we will perform analyses by considering as standard only population of the first year of observation.

Reviewer 3 Report

Dear authors

I have some doubts about how Joinpoint regression has been applied. Generally, this type of regression is used with incidence rates or mortality rates. Considering that all cases are new, it could be applied to hospitalization cases associated with HPV-related diseases. In addition, the Jopinpoint regression is used to detect changes in trends. Has this regression been applied to detect trend changes or isn’t there any?

Figures 1 and 2 are very illustrative. Why not do the same with other diseases? Knowing what has happened in each age group would improve the discussion.

All these considerations are due to the fact that the results of a Joinpoint regression have not been properly exposed. In a Joinpoint regression we should be able to observe the differences by age group, sex and period as well as analyze the temporal trend and changes in trends through the APC.

The authors own the data and have used the joinpoint regression, so showing the APCs for each age group and sex, by each period and for the whole period should be possible. Showing the data in the proper way would improve the discussion section.

Table 1 should add the SD of each rate.

The title of figure 1 is not well defined. The graph shows the adjustment made by the regression and not the APC, the APC for all period is simply a value.

Author Response

Reviewer #3:

Dear authors

I have some doubts about how Joinpoint regression has been applied. Generally, this type of regression is used with incidence rates or mortality rates. Considering that all cases are new, it could be applied to hospitalization cases associated with HPV-related diseases. In addition, the Jopinpoint regression is used to detect changes in trends. Has this regression been applied to detect trend changes or isn’t there any?

Figures 1 and 2 are very illustrative. Why not do the same with other diseases? Knowing what has happened in each age group would improve the discussion.

All these considerations are due to the fact that the results of a Joinpoint regression have not been properly exposed. In a Joinpoint regression we should be able to observe the differences by age group, sex and period as well as analyze the temporal trend and changes in trends through the APC.

The authors own the data and have used the joinpoint regression, so showing the APCs for each age group and sex, by each period and for the whole period should be possible. Showing the data in the proper way would improve the discussion section.

Table 1 should add the SD of each rate.

The title of figure 1 is not well defined. The graph shows the adjustment made by the regression and not the APC, the APC for all period is simply a value.

Answer:

Dear Reviewer,

Thank you for revising our manuscript and for your suggestions and comments that allow us to improve the quality of the manuscript. Below you will find a point by point answer to your questions.

A: I have some doubts about how Joinpoint regression has been applied. Generally, this type of regression is used with incidence rates or mortality rates. Considering that all cases are new, it could be applied to hospitalization cases associated with HPV-related diseases. In addition, the Jopinpoint regression is used to detect changes in trends. Has this regression been applied to detect trend changes or isn’t there any?

Q: As you have stated, we used Joinpoint regression analysis in order to detect trends of HPV related hospitalizations. A similar approach has been carried out in other similar studies (Baldo et al. BMC Infect Dis. 2013; 13: 462, Kuhdari et al. J Public Health (Oxf). 2017 Dec 1;39(4):730-737) that evaluated hospitalization rates. Our analyses suggest that all investigated diseases showed statistically significantly declining trends with APC ranging from -3.73% (anal cancer) and -12.51% (genital warts). Overall, significant trends over the considered years were assessed as average annual percent changes (APC) that we used as a summary measure of the trend over a given fixed interval. We computed APC  as a weighted average of the annual percent change emerging from the joinpoint model, using weights equating to the length of the period interval (Kim et al. Permutation tests for joinpoint regression with applications to cancer rates. Stat Med. 2000;19:335–51). Moreover, the following statements were added to the Discussion section: “in our statistical analysis we reported only APC omitting to describe changes in trends during the specified fixed time interval. The reason for a such choice was due to the fact that, when different intervals trends were identified over the period, the slope coefficients for segment in the range of years were relatively similar. Moreover, we were interested into identify a single summary measure over the entire interval time period for each disease”.

We added this statement in the statistical analysis of the material and methods section.

Q: Figures 1 and 2 are very illustrative. Why not do the same with other diseases? Knowing what has happened in each age group would improve the discussion.

All these considerations are due to the fact that the results of a Joinpoint regression have not been properly exposed. In a Joinpoint regression we should be able to observe the differences by age group, sex and period as well as analyze the temporal trend and changes in trends through the APC.

A: We agree that Figure 1 and 2 are very informative and they well support discussion especially for cervical cancers. In a first moment, we have also produced APC figures for the other diseases but we did not add these last figures to the manuscript for three main reasons. Firstly, the choice of stratifying by age the APC cervical cancer was supported by the fact that this is the only pathology for which, to date, there are evidence based primary and secondary preventive strategies that are offered at two different age groups (vaccination at 12 years and screening at 25-64 years). Secondly, by adding some other figures the manuscript would be characterized by a very high number of figures and also results and discussion would be probably too rich and long. Ultimately, the other diseases show relatively small hospitalization rates in our population and thus it could be less useful to explore them in details. According to your comments, we have tried to improve results and discussion in order to better explain the findings obtained by the Jointpoint regression analysis. However, if required from you or requested by the Editor, we will reconsider our preference by adding some more figures including all investigated diseases, stratified by age and sex.

Q: Table 1 should add the SD of each rate.

A: We have added SD to each hospitalization rates.

Q: The title of figure 1 is not well defined. The graph shows the adjustment made by the regression and not the APC, the APC for all period is simply a value.

A: We apologize for this inaccuracy. The title has been rewritten according to your consideration.

Round 2

Reviewer 3 Report

I have no comments